# The Out-of-Plane Compression Behavior of Cross-Ply AS4/PEEK Thermoplastic Composite Laminates at High Strain Rates

**DOI:** 10.3390/ma11112312

**Published:** 2018-11-17

**Authors:** Huiran Zou, Weilong Yin, Chaocan Cai, Bing Wang, Ankang Liu, Zhen Yang, Yibin Li, Xiaodong He

**Affiliations:** Center for Composite Materials and Structures, Harbin Institute of Technology, Harbin 150080, China; zouhr2015@gmail.com (H.Z.); C39008@163.com (C.C.); wangbing86@hit.edu.cn (B.W.); liuankang1993@126.com (A.L.); zhen940729@163.com (Z.Y.); Hexd@hit.edu.cn (X.H.)

**Keywords:** AS4/PEEK laminates, high strain rate, compression properties, thermoplastic, SHPB test

## Abstract

The dynamic mechanical behavior of thermoplastic composites over a wide range of strain rates has become an important research topic for extreme environmental survivability in the fields of military protection, aircraft safety, and aerospace engineering. However, the dynamic compression response in the out-of-plane direction, which is one of the most important loading conditions resulting in the damage of composite materials, has not been investigated thoroughly when compared to in-plane compression and tensile behavior under high strain rates. Thus, we used split Hopkinson pressure bar (SHPB) tests to conduct the out-of-plane compression test of cross-ply carbon fiber-reinforced polyetheretherketone (AS4/PEEK) composite laminates. Afterward, the damage mechanism under different strain rates was characterized by the macrostructure morphologies and scanning electron microscope micrographs. Two major cases of the incomplete failure condition and complete failure condition were discussed. Dynamic stress-strain curves expound the strain rates dependencies of elastic modulus, failure strength, and failure strain. An obvious spring-back process could be observed under incomplete failure tests. For the complete failure tests, secondary loading could be observed by reconstructing and comparing the dynamic response history. Lastly, various failure modes that occurred in different loading strain rates illustrate that the damage mechanism also shows obvious strain rate sensitivity.

## 1. Introduction

The growing application of fiber-reinforced polymer composites demonstrates their vast advantages and potential when compared to conventional materials. It is noted that using these under extreme conditions leads to a more comprehensive demand for mechanical properties instead of solely the desire for higher specific strength and specific modulus. The dynamic mechanical behavior of composites over a wide range of strain rates has become an important research topic for impact resistant structure design in the fields of military protection, aircraft safety, and aerospace engineering. The mechanical response and failure modes of fiber-reinforced composites are significantly influenced by the matrix type. Many investigations have illustrated that thermoset composites will undergo more extensive damage under low-velocity impact, which can seriously weaken the residual strength and stiffness properties [1,2]. Thermoplastic composites such as the carbon fiber-reinforced PEEK evaluated in Bishop’s research [3] generally exhibit better impact resistance because of higher damage tolerance. Dorey et al. [4] and Ishikawa et al. [5] confirmed this viewpoint by comparing the extent of impact damage between carbon fiber-reinforced PEEK and carbon fiber-reinforced epoxy. Recently, a strain-rate-dependent damage model was proposed by Wang et al. [6] to accurately predict the low-velocity-impact-induced damage evolution of composite laminates and Thierry et al. [7] presented a multi-scale wave propagation model for two-dimensional periodic textile composites. In contrast, the properties of rate-dependent mechanical behavior and failure mechanisms under high strain rates are not thoroughly investigated. Few researchers have made effort on this aspect and pointed out the significant value of this research for advanced composite materials and structures exposed to high-velocity dynamic loadings [8]. Therefore, in this paper, we have attempted to insightfully discuss the response of cross-ply AS4/PEEK thermoplastic composite laminates at high strain rates and explore its damage evolution and failure mechanism under extreme conditions.

The split Hopkinson bar has always been employed in dynamic mechanical property tests for various materials. Khosravani et al. [9] and Weinberg et al. [10] have recently employed this apparatus in the study of the impact resistance of ultra-high-performance concrete. Simultaneously, it has also been one of the most widely used apparatuses for characterizing the dynamic behavior of composite materials in the range of 100–10,000/s strain rates [11]. Ninan et al. [12] used a split Hopkinson pressure bar to characterize the high strain rate behavior of fiber composite off-axis specimens. Through experimental and numerical methods, they analyzed the effect of various factors on test accuracy such as specimen-bar interface friction, extension-shear coupling, and the rise time of the loading pulse. Chen et al. [13] investigated the tensile and compressive properties of 0°/90° woven carbon fiber reinforced epoxy composites from 10^−3^ to 10^3^ strain rates by using a split Hopkinson tension/pressure Bar. Both the tensile and compressive stress-strain curves were acquired under in-plane loading along the 0° direction. In comparison to the test results of coupon and block specimens, Tsai et al. [14] chose S2/8552 glass-epoxy composite block off-axis specimen as the uniform standard specimen in a low strain rate to a high strain rate compression test. A constitutive model for a high strain rate response was developed under a relatively low strain rate and then verified through a SHPB test where the loading strain rates were as high as 700/s.

So far, most research has been based on the investigation of in-plane compression performance. As early as 1986, Kumar et al. [15] investigated the dynamic compressive behavior of unidirectional glass-epoxy composite at an average strain rate of 265/s for different fiber orientations with respect to the loading axis. The results indicated that this material was strain-rate-sensitive for all fiber orientations, but the dynamic ultimate strength and failure mode varies with the fiber direction. Hosur et al. [16] also studied the off-axis high strain rate compression response of stain weave carbon/epoxy composites. The effect of off-axes angles to strain rate sensitivity, failure modes, and peak stress were evaluated. Li et al. [17] conducted an experimental study on the high strain rate compression of 3D-multi-axial warp knitted carbon/epoxy composites. The effects of different fiber architectures and temperatures on the in-plane dynamic compression properties of thermoset composites were discussed in their study.

Little research has been done on the out-of-plane compression performance of composite materials under high strain rates [18]. The relatively representative work is done by Tarfaoui et al. [19]. They not only studied the influence of fiber direction on the dynamic compression mechanical properties in an out-of-plane direction but also summarized the differences of their damage modes. Recently, Arbaoui et al. [20] studied the high strain rate compression mechanical properties of E-glass/vinylester laminated composites in both in-plane and out-of-plane directions. Additionally, thermoplastic-based composite laminates made from Kevlar fiber and maleic anhydride grafted-PP (MAg-PP) matrix were reported in the investigation on a high strain rate compression response presented by Kapoor et al. [21]. Their research on the out-of-plane compression properties and failure analysis of thermoplastic composites were an important reference for our study.

For AS4/PEEK composites, some of the most influential work performed has been presented in the following literature. Weeks and Sun [22] investigated the rate-dependent behavior of an AS4/PEEK thermoplastic composite from 1 × 10^−6^ to 10^3^/s and introduced models to describe its non-linear behavior. Kawai et al. [23] conducted a micromechanical analysis on the off-axis rate-dependent inelastic behavior of unidirectional AS4/PEEK at high temperatures. Totry et al. [24] predicted the failure locus of a unidirectional AS4/PEEK by numerical simulation under the same loading type described in Vogler and Kyriakides’s [25,26] study. The main deformation and failure modes shown in the experiment and prediction results under transverse compression and longitudinal shear are in good agreement. Additionally, in Dong’s et al. research [27], the strain rate effects of unidirectional thermoset composites (carbon/epoxy) and unidirectional thermoplastic composites (carbon/PEEK) were all involved. However, the difference is that they focus on the sensitivity of interlaminar shear performance to strain rate. They also found that the interlaminar shear strength of carbon fiber reinforced epoxy is weakly related to the strain rate while carbon fiber-reinforced PEEK is almost not related to strain rate. As a thermoplastic composite with better impact resistance, their out-of-plane compression mechanical properties and failure mechanism at high strain rates were not sufficiently investigated.

Consequently, this paper mainly attempts to supplement the scarcity of understanding on the dynamic compression mechanical properties of cross-ply thermoplastic laminate composites in the out-of-plane direction. The stress–strain response of AS4/PEEK composite laminates at different strain rates will be presented to clarify the strain rate dependence of out-of-plane compression behavior. The damage kinetics of this material, according to two major cases of the incomplete failure condition and complete failure condition, will also be described. The macrostructure morphologies and scanning electron microscope micrographs will be used to characterize the failure mode and damage mechanism of specimens during the loading failure process. Various forms of failure mechanisms at different strain rates were presented in order to summarize the strain rate sensitivity failure behavior of this material.

## 2. Materials and Methods

### 2.1. Material and Specimens

The material tested in this study was an AS4/PEEK composite laminate with a uniform stacking sequence of (0°/90°)_4s_. The TenCate Cetex® TC1200 PEEK resin system was reinforced by AS4 carbon fibers in the unidirectional prepregs. These prepregs have a thickness of 0.125 mm and the fiber volume fraction is 59%. The thickness of cross-ply AS4/PEEK composite laminates is 2 mm. They were made by using the autoclave process. A total number of 16 layers of unidirectional prepregs were laid up and cured by the recommended process [28]. It should be noted that individual layers with the desired orientation were vacuum bagged and the vacuum should be maintained throughout the entire process. Then they were placed in the autoclave and heated to about 700 to 750 °F (370–400 °C) while the pressure was increased from ambient to 100–300 psi (7–21 bar) for about 5 to 30 min. The part was then cooled to room temperature at a cooling rate of 5 to 20 °C. This material has a very low void content (<1%) and low moisture absorption. Good impact resistance and excellent structural performance have also promoted the application of carbon fiber reinforced thermoplastic resin matrix composites.

Composite block specimens were cut from the laminated panels using the waterjet cutting machine. As shown in Figure 1, these block specimens are suitable for the SHPB high strain rate test in both in-plane and out-of-plane direction compression [16]. The mechanical properties of single layer CETEX TC1200 PEEK AS4 are provided by TenCate Advanced Composites (Morgan Hill, CA, USA). They are *E*_1_ = 130 GPa, *E*_2_ = 10 GPa, *G*_12_ = 5.2 GPa, and *ν*_12_ = 0.33. The specifications of specimens and the corresponding loading strain rates utilized in our experiments are presented in Table 1.

### 2.2. Theory and Experimental Scheme

A schematic diagram of a conventional SHPB apparatus can be seen in Figure 2a. It mainly consists of a striker bar, an incident bar, a transmitted bar, a gas gun, and energy absorption system [29]. The specimen is always sandwiched between the incident bar and the transmitted bar after Kolsky first designed this modification based on the original Hopkinson bar [30]. The electrical method for measuring stress pulses was introduced by Daives [31]. He also initially discussed the dispersion effect of the stress pulse during propagation in the pressure bars. In addition, it must be ensured that all pressure bars and specimen are coaxial during the test. The striker bar is launched from the gas gun due to the expansion of high-pressure gas and it impacts the free end of the incident bar at a certain velocity. As described in Chen’s paper [32], a compressive stress pulse is generated in the incident bar and propagates along the bar towards the specimen. Any desired magnitude of stress pulses that does not exceed the elastic limit of the pressure bar is available because the striker bar can be launched at different velocities by adjusting the pressure of the gas gun. When this compression pulse propagates to the specimen-incident bar interface, a portion of it is reflected back to the incident bar due to an impedance mismatch and another portion is transmitted through the specimen into the transmitted bar. The incident and reflected stress pulses can be measured by a strain gauge mounted on the middle of the incident bar and the transmitted wave is measured by a strain gauge mounted on the middle of the transmitted bar, which is shown in Figure 2a. These stress pulses can be used to derive the strain rate ${\dot{\varepsilon }}_{s}\left(t\right)$, strain $\varepsilon_{s}\left(t\right)$, and stress $\sigma_{s}\left(t\right)$ experienced by the specimen during dynamic compression. In order to minimize the inertia effect, the slenderness ratio of specimens is generally between 0.4 and 0.6. Assuming that the length of the specimen is *l_s_*, the cross-sectional area is *A_s_* and the cross-sectional area of the pressure bar is *A_b_*. Based on the one-dimensional stress wave theory, the sample response can be described by the following equations [33,34].
(1)$${\dot{\varepsilon }}_{s}\left(t\right)=\frac{c_{b}}{l_{s}}\left[\varepsilon_{I}\left(t\right)-\varepsilon_{R}\left(t\right)-\varepsilon_{T}\left(t\right)\right]\;$$
(2)$$\varepsilon_{s}\left(t\right)=\frac{-2c_{b}}{l_{s}}\int_{0}^{t}\left[\varepsilon_{I}\left(t\right)-\varepsilon_{R}\left(t\right)-\varepsilon_{T}\left(t\right)\right]dt\;$$
(3)$$\sigma_{s}\left(t\right)=\frac{A_{b}E_{b}}{2A_{s}}\left[\varepsilon_{I}\left(t\right)+\varepsilon_{R}\left(t\right)+\varepsilon_{T}\left(t\right)\right]\;$$

It should be noted that the cross-sectional area of the specimen *A_s_* must always be smaller than that of pressure bar *A_b_* in order to not affect the effectiveness of the experiment. The *E_b_* and *c_b_* involved in the above formulas are intrinsic properties of the pressure bar, i.e., elastic modulus and wave velocity, respectively. If the specimen is in the dynamic stress equilibrium, we obtain the following equation.
(4)$$\varepsilon_{I}\left(t\right)+\varepsilon_{R}\left(t\right)=\varepsilon_{T}\left(t\right)\;$$

Equations (1)–(3) can be simplified by the following formulas.
(5)$$\dot{\varepsilon }=-2\frac{c_{b}}{l_{s}}\varepsilon_{R}\left(t\right)\;$$
(6)$$\varepsilon_{s}\left(t\right)=\frac{-2c_{b}}{l_{s}}\int_{0}^{t}\varepsilon_{R}dt\;$$
(7)$$\sigma_{s}\left(t\right)=\frac{A_{b}E_{b}}{A_{s}}\varepsilon_{T}\left(t\right)\;$$
where *ε_I_*(*t*), *ε_R_*(*t*), and *ε_T_*(*t*) are incident, reflected, and transmitted strain pulses recorded by the strain gauges, respectively.

The SHPB device used in this study was provided by the High Velocity Impact Dynamics Laboratory of the Harbin Institute of Technology, as shown in Figure 2b. The striker bar and the remaining pressure bars in the experimental device are both 16 mm in diameter. The length of the incident bar and the transmitted bar are both 1200 mm while the striker bar is 300 mm. When the specimen was prepared, a small amount of Vaseline was applied to the bar-specimen interface to reduce the effect of friction on the effectiveness of the test. If necessary, the speed of the striker bar can be recorded by an infrared speed detector. All bars are made of tool steel with a yield strength of 923 MPa. They can withstand a maximum impact velocity of up to 40 m/s while still maintaining elastic properties. The pressure bars have the same density at *ρ_b_* = 7830 g/cm^3^, which is a Young’s modulus of *E_b_* = 210 GPa and a cross-sectional area of *A_b_*. The wave speed in the pressure bars is taken as $c_{b}=\sqrt{E_{b}/\rho_{b}}$. Voltage signals measured by strain gages mounted at the midpoint of pressure bars were sent to the oscilloscope through the amplifier and stored in the computer for the calculation of the strain pulses. All experiments conducted in this paper use a data sampling frequency of 12.5 MHz.

The macroscopic morphology of specimens before the test is shown in Figure 2c. After the specimen had been installed, each specimen was loaded at different impact speeds by adjusting the pressure of the gas gun. Therefore, we can obtain the dynamic compression mechanical properties of the specimen under different strain rates. The loading direction of the pressure bar is, as shown in Figure 2d, along the out-of-plane direction of the specimen. By comparing the out-of-plane compressive stress–strain curves at different strain rates, we can obtain the strain rate-dependent material properties of AS4 reinforced PEEK composites. Two representative cases of complete failure and incomplete failure of a specimen will be discussed and summarized in subsequent chapters. The macroscopic morphology of specimens after a test will be used to illustrate the differences in failure modes of this material at different strain rates and strains. In addition, the microscopic morphology characterized by scanning electron microscopy further clarifies the microscopic mechanism of failure of our materials under different test conditions.

## 3. Results and Discussion

With the change of impact velocity, specimens undergo five different strain rates of loading from low to high. When the amplitude of the stress pulse is higher than the ultimate strength of the tested material, the specimen undergoes complete failure and loses the load carrying capacity. Conversely, when the amplitude of the incident stress wave does not exceed the ultimate strength of the material, no damage or only a small amount of damage occurred in the specimen. The material still maintains a certain degree of carrying capacity. In both cases, the strain rate dependence of the material’s compression response is different. The voltage signals of typical incident, reflected, and transmitted waves are shown in Figure 3. As evident in this figure, when the strain rate is 1469/s and 1489/s, the stresses of the incident loading pulses are less than the ultimate strength of the specimen and they do not undergo complete failure during the test, as shown in Figure 3a. When the stresses of the incident pulses are greater than the ultimate strength of the specimen, which is shown in Figure 3b, the specimens will undergo complete failure and lose their carrying load capacity. Comparing these two different cases, both the reflected pulses and the transmitted pulses exhibit different waveform characteristics.

### 3.1. The Strain Rate Dependence of an Out-of-Plane Compression Response

The SHPB graphical data analysis tool developed by Francis et al. [35] was adopted to derive the stress-strain and strain rate-strain curves. We used the one-wave method to analyze experiment data. We can convert the measured voltage signal into a strain pulse using the voltage-to-strain conversion factor. Then the strain rate history can be obtained, according to Equation (5). The strain history and stress history of specimen experienced during the test can be calculated by Equations (6) and (7). These results are presented in Figure 4 and Figure 5, which is classified according to incomplete failure and complete failure. As shown in Figure 4a, it is obvious that the stress-strain responses have the feature of spring back under strain rates of 1469/s and 1489/s. The maximum stress experienced by the specimen and the corresponding maximum strain increase with the rise of the incident stress pulse. It can also be noted that the elastic modulus of the specimen did not change significantly when the specimen deformed within the elastic range. As the amplitude of the stress pulse continued to increase, the specimen began to fail and did not spring back when the strain rate increased to 1832/s. The maximum stress at this time represents the failure strength of the material at this strain rate and the corresponding strain is the failure strain. The stress-strain response indicates that the failure strain of AS4/PEEK thermoplastic composite laminates is as high as 11.3% while the failure strength is 434 MPa. From the strain rates-strain curves shown in Figure 4b, we can further clarify the spring back phenomenon of this material in the incomplete failure tests. When the stress of the incident pulse reaches the failure strength of the test material at a strain rate of 1832/s, the evolution of the corresponding strain rate also indicates that the material will not spring back due to failure.

The corresponding loading pulse with a strain rate of 1832/s can be regarded as the critical stress that caused the complete failure of the test material. Complete failure occurred for all specimens when the incident pulse continued to increase. It can be seen from Figure 5a that the failure strength and strain decreased at first and then increased with the increase of the strain rate from 1832 to 6848/s. Jamal Arbaoui et al. [17] has also obtained a similar result in their out-of-plane dynamic compression tests. The decrease of failure strength at a strain rate of 3866/s is due to the following reasons. First, the more serious inherent defects may lead to the decrease of strength. Second, the higher the strain rate, the faster the defect-dominated failure occurs and to greater extents. The initiation of cracks usually begins at the inherent defects of the specimens. Since crack propagation through different layers is accompanied by a large number of fiber breaks and matrix failure, more crack branches followed by a main crack can cause the depletion of further carrying capacity when the amplitude of the incident pulse increases significantly. The premature failure of the specimen, which is accompanied by more debris is attributed to the decrease of residual strength after damage. We observed a significant increase in the failure strength only when the strain rate enhancement effect was sufficient to offset the effect of the defect on the strength. It was observed that the failure strength increased 117% from 338 to 734 MPa with an increase of the strain rate from 3866/s to 6848/s. At the same time, the failure strain also increased about 52.9% from 8.5% to 13%. The dynamic compression modulus of the specimen also increased from 3976 to 5646 MPa. As shown in Figure 5a, stress-strain curves at strain rates of 3866/s and 6848/s show that the strain rate effect of material strength is sufficiently obvious and it is not obscured by the differences of material defects. This means that the strain rate sensitivity of failure strength and strain are significant in the characterization of the out-of-plane compression response of (0°/90°)_5_ AS4/PEEK laminates. The dynamic modulus is only significantly affected by the strain rate in complete failure tests. Obvious secondary loading can be observed when the strain rate is greater than 1832/s. The evolution of strain rates presented in Figure 5b are quite different from those in Figure 4b. These strain rates will increase again after the specimens are in failure and will not decrease below zero when unloading. Both evolutions of the two cases will be described in more detail in the next section.

### 3.2. Damage Kinetics

In order to reconstruct a history of dynamic compression response experienced in the two different situations, we summarized their loading and damage evolution processes. First, a noticeable comparison of the history of stress and strain rates is shown in Figure 6 for the case of incomplete damage. It is easily observed that the whole loading process goes through three stages: compacting, loading, and unloading. During the stage of compaction, there is a little increase in stress while the strain rate increases dramatically to the maximum amount. Subsequently, the specimen experienced the loading stage in which the stress begins to increase with the rapid decrease of the strain rate. This stage continues until the stress value reaches a maximum during which the material strength acts to resist deformation. The final stage is unloading. The stress decreases continuously in this period while the strain rate decreases again rapidly until the minimum value is reached. In this stage, the strain rate is negative for most of the time. It means that the specimen is undergoing a spring back process.

In contrast, a different evolution process of stress known as the strain rate is presented in Figure 7, which is obtained from the complete failure test. In this case, the specimen experienced five stages during the whole loading process. The first two phases are also the compacting and loading stages and they had the same characteristics as what was observed in the incomplete failure test. The difference is that the maximum value reached by the stress at this time is the ultimate strength of the material. When the incident stress pulse continues to load, the specimen loses its load carrying capacity due to failure. This results in a decrease of stress in the third stage and the strain rate increases again. Subsequently, the strain rate fluctuates and the stress increases slightly in the fourth stage. This is mainly because the residual debris has rearranged and recovered carrying capacity again to some extent after the fragmentation of the specimen. Lastly, both stress and strain rates decreased to zero without further reduction. This agrees well with the stress–strain response characterized without the spring back in Figure 5.

### 3.3. Failure Mode and Damage Mechanism

For thermoplastic composites studied in this paper, the mechanical properties are not only related to the strain rate but the failure mode and damage mechanism are also affected by the strain rate. In this section, macrostructure photographs and scanning electron microscopy characterization methods will be used to illustrate the differences in the failure mode and damage mechanism at different strain rates. The macroscopic failure morphology of tested specimens is given in Figure 8. Four different strain rates, i.e., 1489, 1832, 3866, and 6848/s were selected to make a comparison study of the failure mechanism. It can be seen from Figure 8a that the macrostructure of specimen A5 remains intact and no clear visible damages are found under the loading strain rate of 1489/s. This is because the specimen always kept in an elastic deformation state during loading when the stress of the incident pulse is lower than its strength limit. It will spring back quickly and recover the original state as soon as unloading. Only a small amount of fiber breakage damage can be found in the SEM micrographs. When the stress of the incident pulse reaches the strength limit, the specimen will begin failure and irreparable damage will occur. It can be considered that the specimen is in a transition stage of incomplete failure and complete failure when the strain rate is 1832/s. As shown in Figure 8b, a separation of delaminated layer blocks and peeling of fragments occurred after specimen failure. Furthermore, a typical corrugated crack is initiated by an interlayer defect. It propagates as a main crack and crosses multiple layers with a certain angle. As the strain rate increases to 3866/s, the stress of the incident pulse is greater than the strength limit of the material. Material failure appears quickly at such high strain rates. Since the duration time of the incident pulse is constant, the residual specimen will continue to experience a second loading after specimen failure. Therefore, we can see in Figure 8c that the specimen is further broken into pieces based on the failure mode in Figure 8b. In addition, the stress of the incident pulse is already much greater than the material’s strength limit at a strain rate of 6848/s. A significantly different failure mode is shown in Figure 8d. It is obvious that both fibers and matrix have broken into powder and a small amount of residue is compressed into pieces.

The micrographs obtained in the scanning electron microscope characterization are presented in Figure 9, Figure 10, Figure 11 and Figure 12 in order to establish deep insights on rate dependency damage mechanisms of these composite laminates. Figure 9a,b show two positions of fiber breakage damages that occurred in the loaded specimen at 1489/s. The red dotted lines 1 to 4 in Figure 9a indicate the four locations where the damage occurred. In addition, the position located by red dotted lines 1 to 2 in Figure 9b show details more clearly about these damages. These have little influence on the mechanical properties of these laminates because these damages are negligible when compared to most well-preserved fibers. Figure 10a is a macroscopic morphology of inter-layer cracking and corrugated crack crossing multi-layers. As indicated by the red dotted line 1 in Figure 10a, this corrugated crack propagated as a main crack at a certain opening angle of 70° and multi-layers are penetrated when the incident pulse is not much higher than the material strength. Moreover, severe delamination failure is indicated by the red dashed lines 2 and 3. It is evident in Figure 10b that the binding fibers in the longitudinal direction are de-bonded and the matrix is a failure. The enlarged images of positions 1 and 2 show more details of these damages. At the same time, the transverse fibers on the failure plane are sheared and broken into fragments, as shown in position 3 of Figure 10b. As the strain rate increases (3866/s), the surface morphology of further fractured fragments is shown in Figure 11a. The tearing fracture of the partial longitudinal fiber bundle is observed while most of them are basically intact. Additionally, more serious fragmentation of the transverse fibers and matrix are presented in position 2 of Figure 11a. These fractured fibers are distributed in a disorderly manner on the failure surface. Furthermore, it can be observed at the edge of the block fragments that multiple tearing between the longitudinal layers occurs across the relatively complete transverse layup, which is shown in Figure 11b. A notable difference can be found at a higher strain rate (6848/s), as shown in Figure 12a. There is no fiber arrangement being observed in a small amount of residual debris and all the broken staple fibers are randomly arranged. Fiber tears and kinks can be observed at the edges of the debris. What can be observed in Figure 12b is that a large amount of fibers on the surface of the debris have shattered. The difference in failure modes and damage characteristics fully demonstrate the strain rate dependence of the failure mechanism for these thermoplastic composite laminates.

## 4. Conclusions

In this work, the out-of-plane compression behavior of cross-ply AS4/PEEK thermoplastic composite laminates at high strain rates was investigated by using a split Hopkinson pressure bar. Two major cases under the incomplete failure condition and complete failure condition are discussed to compare their dynamic mechanical properties and damage mechanism. The experiment results show that the incomplete failure specimens are mainly subjected to elastic deformation during low incident pulse loading. Although it seems intact in its macrostructure, the incomplete failure specimens have a certain degree of damage caused by the defect. It can be concluded from the obtained stress–strain curves that the increase of the strain rate does not make a significant change in the elastic modulus. However, the increase of the incident pulses caused the specimens to withstand greater strain and stress. During unloading, the stress–strain curves have clear spring back characteristics and the greater the strain rate and strain are, the more obvious the spring back will be. There is a small amount of fiber breakage damage in incomplete failure specimens examined by the scanning electron microscope characterization. It was observed that a transitional strain rate existed before complete failure occurred. At this transitional strain rate, the failure strength and strain first decrease and then increase with an increased strain rate. In other words, the failure strength, failure strain, and dynamic modulus should have increased significantly with the increase of the strain rate in complete failure tests except for the subsequent strength reduction zone arising from a strain rate of 1489/s. This strength reduction zone began with the initial occurrence of complete failure and ended when the enhancement effect of the strain rate was sufficient to offset the weakening effect of defect-dominated damage evolution on the strength. In the case of the complete failure condition, the specimens undergo a secondary loading process after failure. As the strain rate increases from 3866/s to 6848/s, the failure strength increased 117% and the failure strain of the specimen increased by about 52.9%. Furthermore, the dynamic compression modulus of the specimen also increases significantly. In addition, the microscopic characterization shows that the macroscopic failure mode and microscopic failure mechanism of this material change significantly with the increase of the strain rate and the strain. Various forms of damage including fiber breakage, matrix failure, crack propagation, delaminating, fragments separation, and the crushing of fibers and matrix were observed by comparing fractographs of the material under out-of-plane compression with varying strain rates. The strain rate dependencies of their mechanical properties and failure mechanism demonstrated in this study is conducive to providing new thoughts on the impact resistance optimization of composites.

## Figures and Tables

**Figure 1 materials-11-02312-f001:**
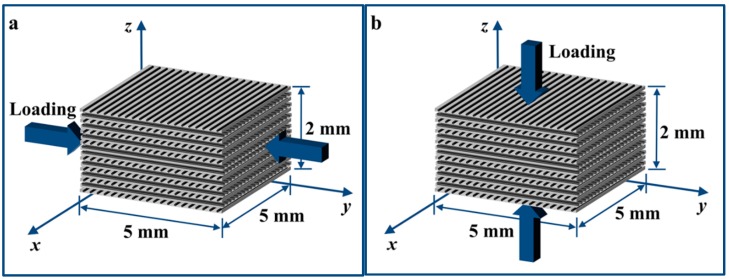
Two typical loading directions in the compression test for block AS4/PEEK composite laminates specimens with a stacking sequence of (0°/90°)_4s_. (**a**) In-plane loading. (**b**) Out-of-plane loading.

**Figure 2 materials-11-02312-f002:**
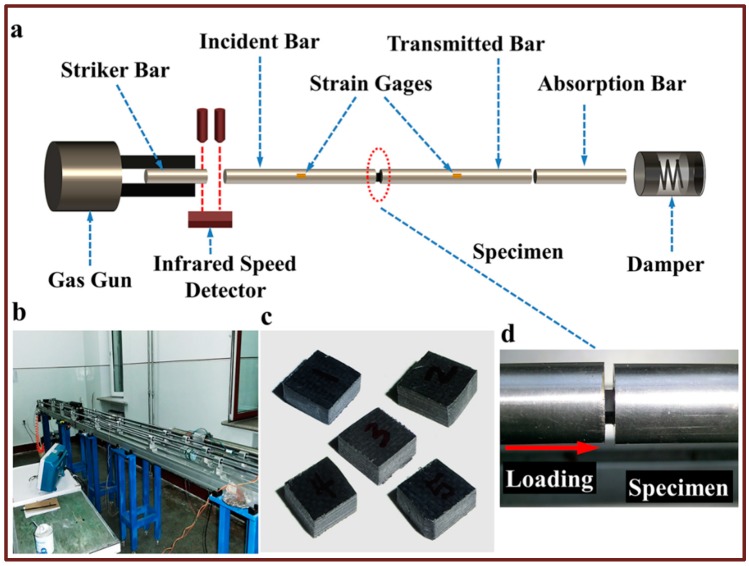
Schematic of a compression split Hopkinson bar apparatus and specimen assembly scheme. (**a**) A schematic diagram of the SHPB apparatus. (**b**) The SHPB setup employed in our tests. (**c**) Geometry of the specimen in compression. (**d**) Position and loading direction of the specimen.

**Figure 3 materials-11-02312-f003:**
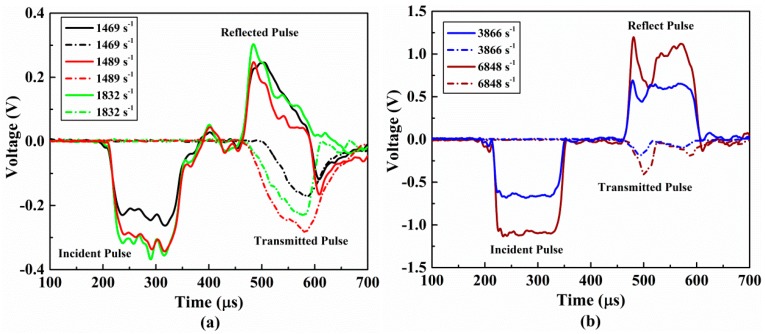
Voltage signals of a typical incident as well as reflected and transmitted pulses obtained from a conventional SHPB test. (**a**) The specimens have not undergo complete failure. (**b**) The specimens have undergone complete failure.

**Figure 4 materials-11-02312-f004:**
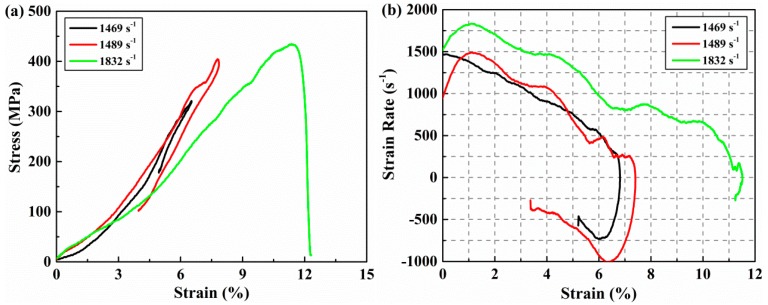
Stress and strain rate as a function of strain at strain rates of 1469/s, 1489/s, and 1832/s during test. (**a**) Stress versus strain. (**b**) Strain rate versus strain.

**Figure 5 materials-11-02312-f005:**
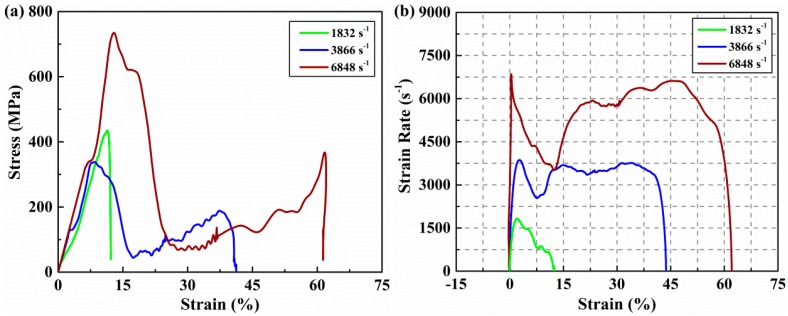
Stress and strain rate as a function of strain at strain rates of 1832, 3866, and 6848/s during the test. (**a**) Stress versus strain. (**b**) Strain rate versus strain.

**Figure 6 materials-11-02312-f006:**
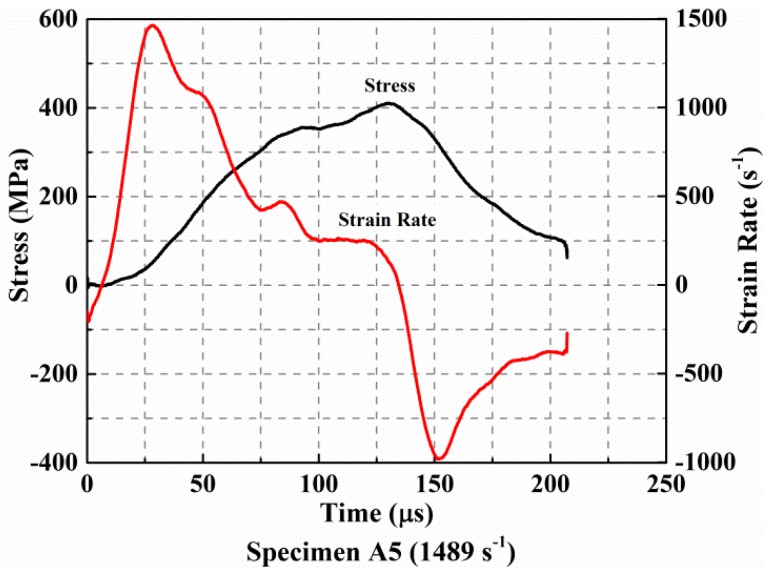
Evolution of the stress and strain rate curves during loading when the specimen has not undergone complete failure.

**Figure 7 materials-11-02312-f007:**
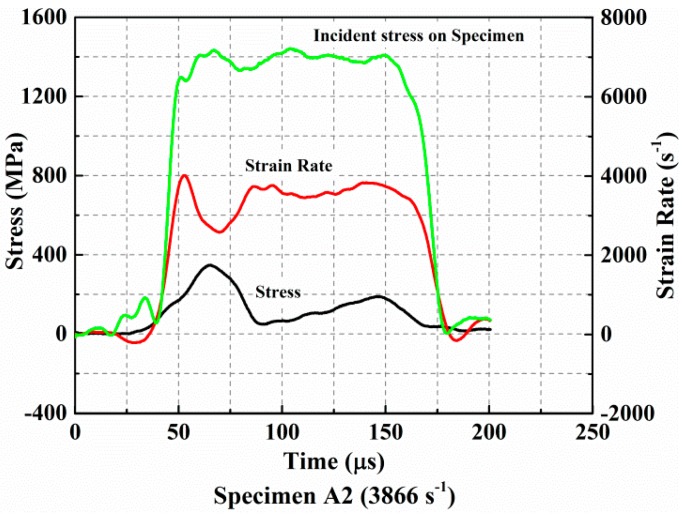
Evolution of the stress and strain rate curves during loading when the specimen has undergone complete failure.

**Figure 8 materials-11-02312-f008:**
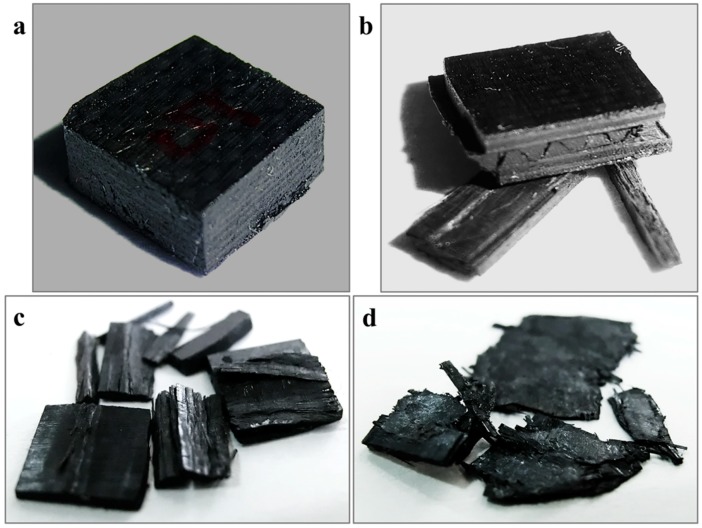
Macrostructure morphologies of (0°/90°)_5_ AS4/PEEK laminated composite after out-of-plane dynamic compression at different strain rates. (**a**) Specimen A5-1489/s. (**b**) Specimen A4-1832/s. (**c**) Specimen A2-3866/s. (**d**) Specimen A1-6848/s.

**Figure 9 materials-11-02312-f009:**
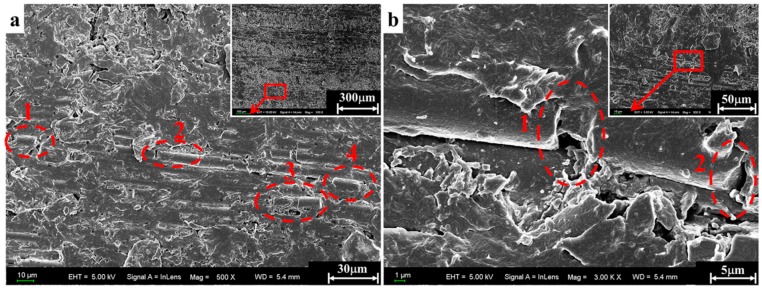
Partial fiber breakage damage observed at different positions of specimen A5 after the test (1489/s). (**a**) The first position with a magnification of 500. (**b**) The second position with a magnification of 3000.

**Figure 10 materials-11-02312-f010:**
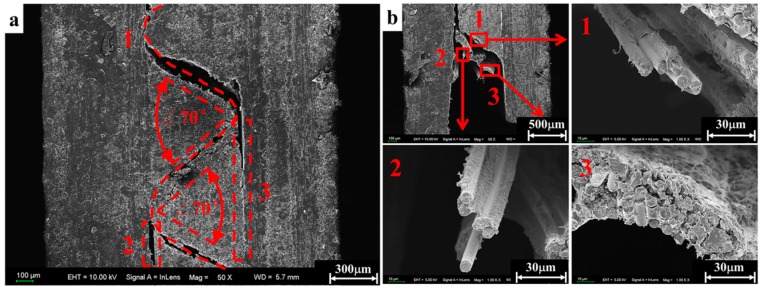
Delaminating and cracking damage occurred at a strain rate of 1832/s. (**a**) Macroscopic appearance of a crack. (**b**) Microscopic failure characteristics.

**Figure 11 materials-11-02312-f011:**
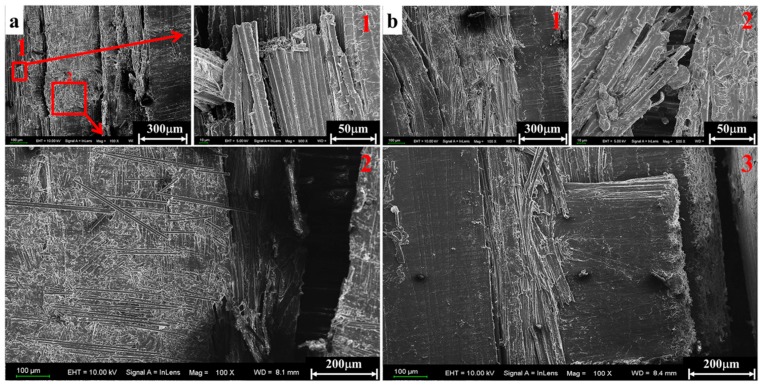
Surface morphology of the debris at a strain rate of 3866/s. (**a**) Failure characteristics of longitudinal and transverse fibers. (**b**) Tearing feature across layers.

**Figure 12 materials-11-02312-f012:**
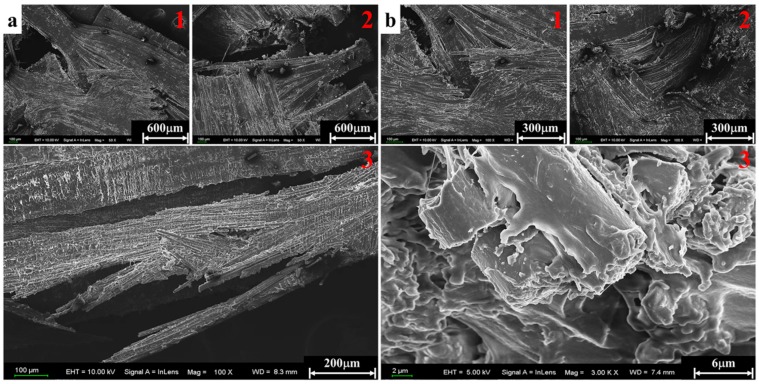
Fibers kink and crush in residual debris at a strain rate of 6848/s. (**a**) Fiber kink. (**b**) Fiber crush.

**Table 1 materials-11-02312-t001:** Loading strain rates and detailed specifications of AS4/PEEK composite laminates specimens.

Specimens	Thickness (mm)	Surface (mm^2^)	Stacking Sequence	Strain Rates (/s)
A1	2.02	4.96 × 4.98	(0°/90°)_4s_	6848
A2	2.00	5.06 × 5.02	(0°/90°)_4s_	3866
A3	1.98	4.98 × 5.04	(0°/90°)_4s_	1469
A4	1.98	5.04 × 5.02	(0°/90°)_4s_	1832
A5	2.02	5.88 × 5.90	(0°/90°)_4s_	1489

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
