# Peer review of "The Out-of-Plane Compression Behavior of Cross-Ply AS4/PEEK Thermoplastic Composite Laminates at High Strain Rates"

_materials, 2018, doi:10.3390/ma11112312_

Round 1

Reviewer 1 Report

The submitted manuscript cannot be accepted for publication in this form, but it has a chance of acceptance after revise and resubmit. My suggestions and comments are as follows:

1. Abstract cover the purpose and results, but give a little information the procedure. The first sentence of the abstract is repeating the title! I think the abstract must be rewrite.

2. In its language layer, the manuscript must be edited by a subject matter expert (there are several grammar mistakes in the text). The general sentences should be removed from introduction.

3. In Section 2.1 it is mentioned that “16 layer unidirectional prepregs were laid up and cured by the recommended process”. In the following paper, this manufacturing process is completely reviewed. So, it is recommend to refer on this paper in Section 2.1: Composite materials manufacturing processes. Appl .Mech. Mater. 2011; 110–116:1361–7.

4. The abbreviations should be introduced for the first time in the text.

5. Several papers are cited in the manuscript, but they are not reviewed. For instance, Ref. [3] to [5] in introduction or [25] to [27] in section 2.2. Each reference should be commented in detail or it should be removed. Moreover, sine this paper deals wave propagation on Hopkinson bar, and also studied a composite material, it seems better to cite some research papers which discuss wave propagation in experimental tests very recently.

6. Since application of SHPB is reviewed newly, it is recommended to consider the following paper and refer to it in section 2.2: Const. Build Mater. 2018; 190-1264-1283.

7. As you mentioned, dynamic equilibrium condition is an important issue on the test. Please explain how you tested this condition to satisfy the Eq. (4).

8. It is better to provide a figure which shows experimental setting completely (instead of schematic). Fig. 2 shows just a small area of the experimental setup. It is recommended to show data acquisition system which should be connected to the SHPB.

9. How did you determine the specimen size?

10. Please present (calculate) how many percent of the loading is transmitted to the transmission bar and how many percent of the impact is reflected? (As you know it is related to the impedance)

11. Please explain why the mentioned strain rates (loading) are considered for the tests?

12. How the curves on Fig. 3 are obtained? How did you extract strain rates from these curves? How many tests are performed for each strain rate? Did you showed the average curve?

13. As you may know, using pulse shaper is necessary in such an experimental test. Please explain if you used a pulse shaper. Otherwise, you should describe how did you obtained the data concerning transmitted and reflected waves.

14. Since stress-Strain and also Strain rate curves are important, description on obtaining the Fig. 4 and Fig. 5 is welcome. Did you present the average curve? (Plot curves with error bar (standard deviation) are welcome).

15. The numbers in Fig. 9 to Fig. 10 are not readable and also are not explained!

16. The conclusion should be reached by the obtained results concerning stress and strain. I believe the conclusion especially in this case should be more accomplish the goals. I think the conclusion must be rewrite.

Author Response

Response to Reviewer 1 Comments

Point 1: Abstract cover the purpose and results, but give a little information the procedure. The first sentence of the abstract is repeating the title! I think the abstract must be rewrite.

Response 1: Thanks for your comment. We have rewritten the abstract according to your suggestions and the relevant requirements in instructions for authors. We added a research background (Refers to Line 14 to Line 16) and highlight the purpose of the study (Refers to Line 16 to Line 19). The procedure of this investigation including the experimental device, materials and damage characterization methods was described in third and fourth sentence (Line 19 to Line 24). A Simplified result discussion and conclusion was presented in the remaining sentences (Line 24 to Line 39). It divides the results into two cases, summarizing their similarities and differences using stress-strain curves and damage kinetics analysis. Damage mechanism was summarized through the failure modes observed in macrostructure morphologies and SEM micrographs of damaged specimens. The detailed results discussion and conclusions were moved to the portion of Results and discussion and Conclusions in the Research Manuscript Sections.

Point 2: In its language layer, the manuscript must be edited by a subject matter expert (there are several grammar mistakes in the text). The general sentences should be removed from introduction.

Response 2: Thanks for this suggestion. We have used the language editing services provided by this journal. Many grammatical errors have been fixed and highlighted in the manuscript. Some general sentences have been modified. The phrase “Generally speaking” has been removed from Line 50, Page 2.

Point 3: In Section 2.1 it is mentioned that “16 layer unidirectional prepregs were laid up and cured by the recommended process”. In the following paper, this manufacturing process is completely reviewed. So, it is recommend to refer on this paper in Section 2.1: Composite materials manufacturing processes. Appl. Mech. Mater. 2011; 110–116:1361–7.

Response 3: Thanks for this suggestion. We have read above reference carefully and cited these papers in section 2.1. The details can be found in Line 142, Page 3 and highlight in the text: “A total number of 16 layers unidirectional prepregs were laid up and cured by the recommended process [28]”

28.    Khosravani, M. R., Composite Materials Manufacturing Processes. Applied Mechanics & Materials 2012, 110-116, 1361-1367.

Point 4: The abbreviations should be introduced for the first time in the text.

Response 4: Thanks for this suggestion. This problem has been revised and the relevant instructions have been added to the corresponding locations and highlighted in the text. The abbreviative “SHPB” has been introduced with the full name of “split Hopkinson pressure bar” when we firstly used them in Line 20 of the abstract. The abbreviative “AS4/PEEK” has been introduced with the full name of “carbon fiber-reinforced polyetheretherketone” when we firstly used them in Line 22 of the abstract. The abbreviative “WMK” has been replaced with the full name of “multi-axial warp knitted” in Ling 95, Page 2. The abbreviative “SEM” has been removed from Line 131, Page 3, and the full name of “scanning electron microscope” was used in our manuscript.

Point 5: Several papers are cited in the manuscript, but they are not reviewed. For instance, Ref. [3] to [5] in introduction or [25] to [27] in section 2.2. Each reference should be commented in detail or it should be removed. Moreover, sine this paper deals wave propagation on Hopkinson bar, and also studied a composite material, it seems better to cite some research papers which discuss wave propagation in experimental tests very recently.

Response 5: Thanks for this suggestion. The Ref. [3] to [5] in the introduction and [25] to [27] in section 2.2 have be commented in detail. The details can be found in Line 55, 56, and 60, Page 2 and highlight in the text: “Thermoplastic composites, such as the carbon fiber-reinforced PEEK evaluated in Bishop’s research [3], generally exhibit better impact resistance because of higher damage tolerance. Dorey et al. [4] and Ishikawa et al. [5] confirmed this viewpoint by comparing the extent of impact damage between carbon fiber-reinforced PEEK and carbon fiber-reinforced epoxy.”

3.      Bishop, S. M., The mechanical performance and impact behaviour of carbon-fibre reinforced PEEK. Composite Structures 1985, 3, (3–4), 295-318.

4.      Dorey, G.; Bishop, S. M.; Curtis, P. T., On the impact performance of carbon fibre laminates with epoxy and PEEK matrices . Composites Science & Technology 1985, 23, (3), 221-237.

5.      Ishikawa, T.; Sugimoto, S.; Matsushima, M.; Hayashi, Y., Some experimental findings in compression-after-impact (CAI) tests of CF/PEEK (APC-2) and conventional CF/epoxy flat plates. Composites Science & Technology 1995, 55, (4), 349-363.

For references 25 to 27, the details can be found in Line 167 to 168, Page 5 and highlight in the text: “The specimen is always sandwiched between the incident bar and the transmitted bar after Kolsky first designed this modification based on the original Hopkinson bar [30]. The electrical method for measuring stress pulses was introduced by Daives [31].” Line 172, Page 5 and highlight in the text: “As described in Chen’s paper [32], a compressive stress pulse is generated in the incident bar and propagates along the bar towards the specimen.”

30.    Kolsky, H., An investigation of the mechanical properties of materials at very high rates of loading. Proceedings of the Physical Society. Section B 1949, 62, (11), 676.

31.    Davies, R., A critical study of the Hopkinson pressure bar. Philosophical Transactions of the Royal Society of London A: Mathematical, Physical and Engineering Sciences 1948, 240, (821), 375-457.

32     Chen, W.; Zhang, B.; Forrestal, M., A split Hopkinson bar technique for low-impedance materials. Experimental mechanics 1999, 39, (2), 81-85.

Point 6: Since application of SHPB is reviewed newly, it is recommended to consider the following paper and refer to it in section 2.2: Const. Build Mater. 2018; 190-1264-1283.

Response 6: Thanks for your suggestion. We have read above reference carefully and cited these papers in section 2.2. The details can be found in Line 166, Page 5 and highlight in the text: “It mainly consists of a striker bar, an incident bar, a transmitted bar, a gas gun and energy absorption system [29].”

29.    Khosravani, M. R.; Weinberg, K., A review on split Hopkinson bar experiments on the dynamic characterisation of concrete. Construction and Building Materials 2018, 190, 1264-1283.

Point 7: As you mentioned, dynamic equilibrium condition is an important issue on the test. Please explain how you tested this condition to satisfy the Eq. (4).

Response 7: Thanks for your comment. We have compared the force at the incident and transmitted bar ends of each specimen. As show in figure S1.a, the force equilibrium condition is meeting the accuracy requirements of our study. In figure S1.b, the force equilibrium condition is achieved after a short time. So we think all of the tests satisfy the Eq. (4).

Figure S1. (a) The forces at the incident and transmitted bar ends of the specimen-A4 (1832/s); (b) The forces at the incident and transmitted bar ends of the specimen-A2 (3866/s)

Point 8: It is better to provide a figure which shows experimental setting completely (instead of schematic). Fig. 2 shows just a small area of the experimental setup. It is recommended to show data acquisition system which should be connected to the SHPB.

Response 8: Thanks for this good suggestion. We have added a picture of the complete experimental setup used in this study, as shown in Figure S3. It is merged into Figure 2 as a Figure 2b.

Figure S2. The SHPB setup employed in our tests.

Figure 2. Schematic of compression split Hopkinson bar apparatus and specimen assembly scheme. (a) A schematic diagram of SHPB apparatus; (b) The SHPB setup employed in our tests. (c) Geometry of specimen in compression. (d) Position and loading direction of specimen.

Point 9: How did you determine the specimen size?

Response 9: Thanks for this comment. The detailed specifications of the specimens are given in Table 1. Their slenderness ratio (l/d) is about 0.4. This design is based on reference [21], and a detailed discussion is given in section 2.1 of this reference. To minimize the inertia effects, the l/d ratio of 0.3 to 0.5 is shown to be good criteria of SHPB specimen design. Considering the low density of composite materials, our slenderness ratio should less than metal specimens. Besides, the cross-sectional area of our specimen is much smaller than that of the pressure bar. This is because we want to achieve the highest possible strain rate. The strength and modulus of the material being tested are much smaller than the pressure bar, so this size will not cause excessive errors.

21.    Kapoor, R.; Pangeni, L.; Bandaru, A. K.; Ahmad, S.; Bhatnagar, N., High strain rate compression response of woven Kevlar reinforced polypropylene composites. Composites Part B 2016, 89, 374-382.

Point 10: Please present (calculate) how many percent of the loading is transmitted to the transmission bar and how many percent of the impact is reflected? (As you know it is related to the impedance)

Response 10: The percentage of the impact reflected to the incident bar can be calculated by the ratio of the reflected strain wave to the incident strain wave area. We chose the test with a strain rate of 1832 as an example. As shown in Figure S3.a, the calculation result obtained by the origin software is 50.4%. The percentage of the loading transmitted to the transmission bar can be calculated by the ratio of the transmitted strain wave to the incident strain wave area. As shown in Figure S3.b, the calculation result obtained by the origin software is 48.2%.

Figure S3. (a) Area ratio of reflected wave to incident wave when the strain rate is 1832/s; (b) Area ratio of transmitted wave to incident wave at a strain rate of 1832/s.

Point 11: Please explain why the mentioned strain rates (loading) are considered for the tests?

Response 11: These strain rates are selected from a set of tests and represent several typical loading cases. For example, strain rates 1469/s and 1489/s represent incomplete failure conditions, 1832/s represents the transitional strain rate before complete failure occurred, and 3866/s and 6848/s represent complete failure conditions. They can fully represent the typical failure modes at different strain rates.

Point 12: How the curves on Fig. 3 are obtained? How did you extract strain rates from these curves? How many tests are performed for each strain rate? Did you showed the average curve?

Response 12: The curves on fig.3 are voltage signals of typical incident, reflected and transmitted pulses collected by the strain gauges. They are amplified and stored in the oscilloscope. Knowing the necessary material and equipment parameters, we can calculate the strain rate from Equation 5. We conducted two groups of experiments after the necessary repetitive experiments. The strain rate was changed by adjusting the pressure of gas gun from low to high, and each group is more than 5 strain rates. The five strain rates given in the paper are selected from one of these groups. Since the pressure of gas gun cannot be precisely controlled, and the shape of the incident wave is affected by accidental factors. It is difficult to obtain the same strain rate in these two groups. The strain rates in the two groups of tests cannot be one-to-one, so the average curve cannot be given. But we have conducted a large number of repetitive experiments before the formal test. Figure 4 is a group of very close repeat experiments.

Figure S4. (a) Typical pulse waveform at a strain rate of 4200/s; (b) The stress-strain curve of the two specimens at a strain rate of 4200/s.

Point 13: As you may know, using pulse shaper is necessary in such an experimental test. Please explain if you used a pulse shaper. Otherwise, you should describe how did you obtained the data concerning transmitted and reflected waves.

Response 13: In this study, we did not use a pulse shaper. In order to achieving constant strain rate loading and stress equilibrium, pulse shapers have been employed in many SHPB tests. After the analysis in Figure S1, the stress equilibrium condition can be easily satisfied during direct loading. In addition, this paper focuses on the qualitative study of strain rate dependency of mechanical property and damage mechanism, rather than accurate strain rate related constitutive relations. So we can mark each test with average strain rate or maximum strain rate. These reasons prompted us to simplify the experiment without using a pulse shaper. However, we do not deny the using of pulse shapers can further improve the accuracy of the experiment.

Point 14: Since stress-Strain and also Strain rate curves are important, description on obtaining the Fig. 4 and Fig. 5 is welcome. Did you present the average curve? (Plot curves with error bar (standard deviation) are welcome).

Response 14: Thanks for this suggestion. We have added a description on obtaining stress-Strain and also Strain rate curves shown in the Fig. 4 and Fig. 5 according to your suggestion. The details can be found in Line 248 to 251, Page 9 and highlight in the text: “We can convert the measured voltage signal into a strain pulse using the voltage-to-strain conversion factor. Then the strain rate history can be obtained according to equation 5. The strain history and stress history of specimen experienced during the test can be calculated by Equation 6 and 7.”

We didn’t use the average curve. The curves presented in our paper are selected from one group of our test. The strain rate in each group of experiments varied from low to high, and we repeated two groups of experiments. Since the pressure of the gas gun is manually adjusted, it is difficult to ensure that the two groups of experiments are performed at a fixed strain rate. A lot of repetitive experiments have been carried out before the formal test, and a fairly perfect set of repetitions is given in Figure S4.

Point 15: The numbers in Fig. 9 to Fig. 10 are not readable and also are not explained!

Response 15: Thanks for this comment. This problem has been revised and the related description has been added to section 3.3.

See details in Line 364, Page 13 “The red dotted lines 1 to 4 in Figure 9a indicate the four locations where the damage occurred. And the position located by red dotted lines 1 to 2 in Figure 9b show details more clearly about these damages.”

Line369, Page 13 “As indicated by  the red dotted line 1 in Figure 10a, This corrugated crack propagated as a main crack at a certain opening angle of 70°, degrees and multi-layers are penetrated when the incident pulse areis not much higher than material strength.”

Line 372, Page 13 “Moreover, severe delamination failure is indicated by the red dashed lines 2 and 3.”

And Line 374, Page 13 “The enlarged images of positions 1 and 2 shown more details of these damages.”

Point 16: The conclusion should be reached by the obtained results concerning stress and strain. I believe the conclusion especially in this case should be more accomplish the goals. I think the conclusion must be rewrite.

Response 16: Thanks for this suggestion. We have rewritten the conclusions according to your suggestions. Some important results concerning stress and strain have been emphasized, the details can be found in Line 456, Page 15 and highlight in the text: “It can be concluded from the obtained stress–strain curves that the increase of the strain rate does not make a significant change in the elastic modulus.”

A detailed description about the characteristics of stress-strain curves from transitional strain rate to high strain rate was added to conclusions, the details can be found in Line 465, Page 15 and highlight in the text: “In other words, the failure strength, failure strain, and dynamic modulus should have increased significantly with the increase of the strain rate in complete failure tests, except for the subsequent strength reduction zone arising from a strain rate of 1489/s. This strength reduction zone began with the initial occurrence of complete failure and ended when the enhancement effect of the strain rate was sufficient to offset the weakening effect of defect-dominated damage evolution on the strength.”

We also added the increments of failure strength and failure strain when they effect by strain rates. The details can be found in Line 472, Page 15 and highlight in the text: “As the strain rate increases from 3866 to 6848/s, the failure strength increased 117%, and failure strain of specimen increased about 52.9%.”

In addition, we have added a description of the significance of our research results to the conclusions. The details can be found in Line 479, Page 15 and highlight in the text “The strain rates dependencies of their mechanical properties and failure mechanism demonstrated in this study is conducive to providing new thoughts on the impact resistance optimization of composites.”

Reviewer 2 Report

In my opinion, the article submitted for review deserves attention. However, some important issues require some explanation:

1. I am asking for clarification and introduction to the article parameters for the production of the material in an autoclave

2. How many samples have been tested for each case?

3. The authors also present in-plane samples in the article. Were the samples tested, compared with out-of-plane?

Author Response

Response to Reviewer 2 Comments

Point 1: I am asking for clarification and introduction to the article parameters for the production of the material in an autoclave

Response 1: Thanks for your comment. We have added the relevant description to section 2.1 according to your request about the article parameters for the production of the material in an autoclave. The added information can be found and highlighted in Line 141 to 145, Page 3 and 4 and highlight in the paper: “It should be noted that individual layers with the desired orientation were vacuum bagged and vacuum should be maintained throughout the entire process. Then they were placed in the autoclave and heated to about 700-750°F (370-400°C) while the pressure was increased from ambient to 100-300 psi (7-21 bar) for about 5-30 minutes. The part was then cooled to room temperature at a cooling rate of 5-20°C.”

2. How many samples have been tested for each case?

Response 2: Thanks for this comment. We conducted two groups of repeated tests. Since the pressure of the gas gun is manually adjusted, it is difficult to ensure that the bullets are launched at the same speed twice. In addition, the pulse waveform will be affected by accidental factors and there is a certain degree of error. Therefore, the strain rate of each test in the same group changes from low to high. But the strain rates in the two groups of tests are not one-to-one correspondence. They all show similar phenomena, so we only select a few representative data from one group. We have done a lot of repetitive testing before the formal test. A set of perfectly repeated tests is given in Figure S4.

Figure S4. (a) Typical pulse waveform at a strain rate of 4200/s; (b) The stress-strain curve of the two specimens at a strain rate of 4200/s.

3. The authors also present in-plane samples in the article. Were the samples tested, compared with out-of-plane?

Response 3: Thanks for your interest in our further research. Yes, after completing the out-of-plane dynamic response study presented in this article, we have performed some dynamic compression tests in the in-plane direction. Some of the experimental results are shown in Figures S5 and S6. As shown in Figure 5, the failure mode of the in-plane dynamic compression test is relatively simple. The most typical form of failure is delamination damage. The number of layers separated after failure increases as the strain rate increases. It can also be seen from Fig. 5 that the mechanical properties are also significantly enhanced with the increase of strain rate. However, since the sample size has been redesigned, the stress balance conditions during the experiment need further discussion. We also did not perform microscopic characterization of fracture morphology. The research work about this part needs to be improved and we will cover this part of the work in detail in subsequent articles.

Figure S5 Macrostructure morphologies of [0°/90°]5 AS4/PEEK laminated composite after in-plane dynamic compression at different strain rates. (a) Specimen 7-831/s. (b) Specimen 5-964/s. (c) Specimen 3-1695/s.

Figure S6 Stress and strain rate as a function of strain at strain rates of 831, 964, and 1695/s during the in-plane dynamic compression tests. (a) Stress versus strain. (b) Strain rate versus strain.